

# Increased mental stress among undergraduate medical students in south-western Saudi Arabia during the COVID-19 pandemic

Nabil J. Awadalla[1,2], Abdullah A. Alsabaani[1], Mohammed A. Alsaleem[1], Safar A. Alsaleem[1], Ayoub A. Alshaikh[1], Suliman H. Al-Fifi[3] and Ahmed A. Mahfouz[1,4]

[1] Department of Family and Community Medicine, College of Medicine, King Khalid University, Abha, Aseer, Saudi Arabia
[2] Department of Community Medicine, College of Medicine, Mansoura University, Mansoura, Egypt
[3] Department of Child Health, College of Medicine, King Khalid University, Abha, Aseer, Saudi Arabia
[4] Department of Epidemiology, High Institute of Public Health, Alexandria University, Alexandria, Egypt

## ABSTRACT

**Background**. The COVID-19 pandemic has disrupted the daily life and academic trajectory of many students. The objectives of this study were to evaluate the effect of the pandemic on perceived stress levels among medical students.

**Methods**. Comparative pre-pandemic and pandemic surveys were conducted among samples of undergraduate medical students. Students responded to a questionnaire including personal and academic data, and Cohen's Perceived Stress Scale (PSS).

**Results**. Overall, the prevalence of high perceived stress during the pandemic (20.6%) was significantly higher ($p = 0.001$) than pre-pandemic (11.6%). A multivariable analysis revealed that the independent factors associated with high perceived stress were: participation in the study during the pandemic (aOR = 1.79, 95% CI: 1.22–2.63), female sex (aOR = 1.74, 95% CI: 1.23–2.47), younger age (aOR = 1.62, 95% CI: 1.04–2.55) and lower family income (aOR = 1.50, 95% CI: 1.12–2.03). PSS score was negatively correlated with increasing age, family income, and academic level. PSS score was positively correlated with: worries about the possible disruption of education or exams, excessive news exposure, worries about the possibility of COVID-19 infection, and the effects of mandatory isolation and social distancing.

**Conclusion**. The COVID-19 pandemic increased the level of stress among medical students. Female students, younger students, and those in lower academic grades are the most at risk of having high stress. Worries about possible academic disruptions due to the pandemic are significant stressors. The implementation of online stress management programs is recommended.

Corresponding author
Ahmed A. Mahfouz,
mahfouz2005@gmail.com

## INTRODUCTION

The SARS-CoV-2 virus, the causative agent for COVID-19, was first observed in China in December 2019, but rapidly spread all over the globe, causing a life-threatening global pandemic (*World Health Organization, 2020*). The Saudi government declared the first in-country case of COVID-19 on the 2nd of March 2020. On the 8th of March 2020, the Saudi Ministry of Education announced the closure of all educational institutions and shifted education from face-to-face to online learning. These decisions were rapidly followed by strict measures for social distancing and a full country lockdown (*Yezli & Khan, 2020*).

Changes in daily habits and routines became compulsory to minimize disease spread. In addition to living with a fear of infection, disease, death, social and family disturbances, educational and income instabilities (*Ahorsu et al., 2020*), decreased life satisfaction and decreased positive emotions were also reported (*Zhang et al., 2020*). Worldwide studies indicate an increase in the level of negative psychological impacts such as high stress, anxiety, and depression, caused by the pandemic; these can all have major impacts on mental health (*Alkhamees et al., 2020*; *Joseph et al., 2021*; *Rogowska, Kuśnierz & Bokszczanin, 2020*).

Although young adults are at low risk of COVID-19 infection from the original strain of the virus (*Vieira et al., 2020*), college students may represent a vulnerable population for mental health problems during the pandemic. Changes in teaching methods, difficulties in delivering clinical training sessions, and the possibility of education disruption due to the pandemic are challenging situations for medical students in addition to the pandemic-related stress faced by the general population (*Aslan & Pekince, 2021*; *Aslan, Ochnik & Çınar, 2020*; *Cao et al., 2020*; *Husky, Kovess-Masfety & Swendsen, 2020*).

In a sample of college students in the United States, over two-thirds of those who reported increases in stress during the pandemic recognized disturbances in educational routine as the most important contributing factor (*Wang et al., 2020*). Also, most college students report problems related to online teaching during the pandemic (*Gillis & Krull, 2020*). Therefore, we hypothesize that COVID-19-related challenges would represent a considerable stressor and result in a significant increase in perceived stress during the pandemic. High stress may lead to unfavorable impacts on the cognitive function of medical students as well as their ability to learn (*Dahlin, Joneborg & Runeson, 2005*).

Studies indicate individual differences in the mental health impact of the pandemic with some characteristics showing more mental health impacts such as being female, being younger, and being relatively more concerned about loved ones contracting the disease (*Aslan & Pekince, 2021*; *Wang et al., 2020*). Therefore, in-depth research and the monitoring of perceived stress levels of medical students during the pandemic are crucial.

Information about the COVID-19-related stressors of medical students, including fearing infection and the physical, mental, social, educational, and economic impacts of that infection, would help the universities improve coping strategies during the pandemic and similar conditions. Studies suggest that coping strategies could be beneficial in mitigating these kinds of stressors, helping reduce stress and its impacts on mental health (*Charles et al., 2021*; *Rogowska, Kuśnierz & Bokszczanin, 2020*).

To our knowledge, none of the research about the impact of the pandemic on perceived stress includes a pre-pandemic comparison group from the same college, so it has been difficult to conclude whether the prevalence of high stress during the pandemic exceeds the pre-pandemic prevalence among similar college students.

This study aimed to better understand the effect of the COVID-19 pandemic on high perceived stress levels among medical students using two-phased research: before and during the pandemic. This study also sought to explore possible pandemic stressors and identify students who are more vulnerable to developing high stress.

## METHODS

### Study design and setting

Comparative pre-pandemic and pandemic surveys were conducted among undergraduate male and female medical students at King Khalid University (K.K.U.). K.K.U. is located in Abha city, in the Aseer region of southwest Saudi Arabia. The university includes 29 colleges with a total of 60,312 male and female students. The study in the college of medicine requires 6 academic years (12 academic levels). Each academic year is divided into two academic levels, each one is followed by a final exam and they are separated with the midyear vacation.

### Study participants and procedure

#### The first survey

The first survey was conducted in the 2018/2019 academic year (prior to the COVID-19 pandemic). A self-administered questionnaire was distributed in person to the study participants by level eight medical students as a part of their training in community medicine courses.

#### The second survey

When the second survey was distributed, university classes had been shifted to online teaching methods due to pandemic control measures. Participants in this survey completed an online Google form survey in the English language developed by the study researchers. The survey link was circulated to class leaders at each academic level from December 2020 to March 2021. Each class leader then distributed the link to all class students through email and the WhatsApp communication platform. Reminders were sent to help increase the survey response rate.

The minimum required sample size was calculated using the Epi info program version 7.2. This calculation used an anticipated prevalence of high perceived stress among students of 33.8% (*Rebello, Kallingappa & Hegde, 2018*), a confidence level of 95%, and an acceptable margin of error of 5% for the first survey and 4% for the second survey. The estimated sample sizes needed were 339 students for the first survey and 524 students for the second survey. Data were collected from 353 and 704 medical students out of 1,250 and 1,400 students, respectively to account for non-response in the first and second surveys. Students included in the first survey were selected using the stratified cluster random sampling technique. Students were stratified by academic level. Within each level a section or group

(cluster) was randomly chosen. All listed students in each group were involved in the study whenever possible. In the second survey, all students in each level were invited until the required sample size was attained.

## Impact of the COVID-19 pandemic in Saudi Arabia on the study

The first part of the study was started and completed in the 2018/2019 academic year, before the COVID-19 pandemic began. After the WHO pandemic announcement, the Saudi Arabian government applied a lockdown in most sectors. On the 9th of March 2020, the Saudi Ministry of Education immediately moved all government and private educational institutions to online teaching as a pandemic control measure. During the second survey, all Saudi universities were operating e-learning platforms using the Blackboard system. In the college of Medicine, all classes, including practical and clinical teaching, were taught online, but all exams were taken in-person at the college with certain COVID-19 mitigation measures in place (*Al-Kadri, Al Moamary & Al Knawy, 2020*). Starting from May 2020, throughout the second survey, the government launched a widespread campaign called ''We All Return Cautiously''. This campaign used established guidelines on the slow and safe return to life before the COVID-19 pandemic. These guidelines included obligatory wearing of face masks in public, measuring people's temperature before entering public places, implementing a maximum capacity for closed locations, ensuring social distancing and hand sanitation. A student identified to have fever was isolated, and local health authorities were notified of the necessary actions (*Sayed, 2021*). Starting from mid-December 2020, the Saudi ministry of health started the national COVID-19 vaccination program.

## Study tool

In the first and second surveys, data were collected using a validated structured questionnaire (*Awadalla, 2019*). The questionnaire included the following items: (a) personal and academic data, including age, sex, family income, smoking status, academic level, and grade point average (G.P.A.), and (b) Cohen's Perceived Stress Scale (P.S.S.) (*Cohen, Kamarck & Mermelstein, 1983*). In the second survey, the questionnaire additionally included eight COVID-19-related questions to explore the pandemic-related stressors.

The P.S.S. Scale is a 10-question self-report tool that measures the perceived stress individuals have been subjected to within the previous month. Each question is ranked on a five-point Likert scale (0 = Never to 4 = Very Often) depending on the degree of how uncontrollable, unpredictable, and overloaded subjects perceive their lives to be, as well as their present degree of stress. This tool has been shown to have good psychometric properties (*Lee, 2012*). The P.S.S. scores were grouped into low, moderate, and high stress groups, according to *Cohen, Kamarck & Mermelstein (1983)*: scores from 0 to 13 were considered low stress, scores from 14 to 26 were considered moderate stress, and scores from 27 to 40 were considered high perceived stress.

The eight COVID-19 related questions included in the questionnaire were written in a 3-point Likert scale structure (0 = Never to 2 = Very Often) designed to assess the degree of worry and concern the respondents had about the effect of the COVID-19 pandemic,

associated governmental measures, and potential COVID-19 infection on health and life. The researchers developed these questions after an extensive review of the literature and a panel of four experts in the Community Medicine department, K.K.U., validated the questions. The content validity ratio for the 8 questions ranged from 0.92-to−0.96. Cronbach's alpha was 0.86 indicating internal consistency reliability of the scale.

## Ethical approval

The Ethical Committee of Scientific Research, King Khalid University, approved the first (ECM#2019-23) and the second (ECM#2020-1205) surveys. The participants provided their written informed consent to participate in this study.

## Data collection

Participation in the study was completely voluntarily. Data confidentiality and students' privacy were assured. The study was conducted according to the ethical codes of the relevant national and institutional committees on human research and the Helsinki Declaration of 1975, as revised in 2008. All incomplete questionnaires were eliminated.

## Data analysis

Data were entered, double-checked, and analyzed using the IBM SPSS Statistics version 22 software package (I.B.M. Corp., Armonk, NY, USA). Chi square test of independence was done to compare the personal, academic data, and level of perceived stress of the students before and during the COVID-19 pandemic. Student's $t$-test was used to compare the average P.S.S. score between male and female students. A multivariable logistic regression analysis model was used to assess the independent factors of high perceived stress in the whole sample. Adjusted odds ratios (aOR) and 95% confidence intervals (95% C.I.s) were calculated to explore the significant factors. The outcome variable was perceived stress (high perceived stress vs. low to moderate perceived stress). The following predictors of high perceived stress were dichotomized variables: sex (female vs. male), age in years ($\leq 22$ vs. $\geq 23$), smoking status (smoker vs. non-smoker) and the timing of the study (during the pandemic vs. before the pandemic). Income was a 3- level variable as per Table 1, with 'Sufficient and exceed' as the reference category. Spearman's rank correlations were used to correlate students' personal characteristics (age in years, family income ranked from insufficient to sufficient and exceeds), academic data (students' educational level and GPA score), and responses to the eight COVID-19 related questions (responses ranged from never to very often) with the overall score of P.S.S. scale. $P$-values of less than 0.05 were considered statistically significant.

# RESULTS

## Description of the study population

Table 1 describes the personal and academic characteristics of the medical students who participated in the study before and during the COVID-19 pandemic. Three hundred and fifty-three medical students participated in the pre-COVID-19 pandemic part of the study, and 704 students completed the online questionnaire during the pandemic. The two

**Table 1 Personal and academic characteristics of medical students included in the study both before and during the COVID-19 pandemic.**

| Factors | Overall N=1057 n (%) | Before pandemic N=353 n (%) | During the pandemic N=704 n (%) | $\chi^2$ | P-value |
|---|---|---|---|---|---|
| Age (years) | | | | | |
| $\geq 23$ | 826 (78.1) | 270 (76.5) | 556 (79.0) | 0.854 | 0.385 |
| $\leq 22$ | 231 (21.9) | 83 (23.5) | 148 (21.0) | | |
| Sex | | | | | |
| Male | 526 (49.8) | 173 (49.0) | 353 (50.1) | 0.121 | 0.754 |
| Female | 531(50.2) | 180(51.0) | 351 (49.9) | | |
| Family income | | | | | |
| Sufficient and exceed | 679(64.4) | 273 (77.8) | 406 (57.7) | | |
| Sufficient | 348 (33.0) | 74 (21.1) | 274 (38.9) | 41.850 | 0.001 |
| Insufficient | 28 (2.7) | 4 (1.1) | 24 (3.4) | | |
| Smoking status | | | | | |
| Current smoker | 124 (11.7) | 40(11.3) | 84 (11.9) | 0.820 | 0.840 |
| Non-smoker | 933 (88.3) | 313 (88.7) | 620 (88.1) | | |
| Academic level | | | | | |
| 1–6 | 318 (30.1) | 110(31.2) | 208 (29.5) | 0.292 | 0.619 |
| 7–12 | 739 (69.9) | 243 (68.8) | 496 (70.5) | | |
| GPA[a] | | | | | |
| $\leq 2.5$ | 23 (2.5) | 7 (3.1) | 16 (2.3) | 0.433 | 0.472 |
| +2.5 | 908 (97.5) | 222 (96.9) | 686 (97.7) | | |

Notes.
$\chi^2$, Chi-square test; GPA, Grade point average.
[a] Missing data for 106 students.

groups were matched on age, sex, academic level, and G.P.A. score ($p > 0.05$). However, the family income of the two groups was statistically significantly different. Reported family income levels of just sufficient or insufficient were significantly ($p = 0.001$) lower among participants during the pre-pandemic period (21.1% and 1.1%, respectively) compared to students who participated during the pandemic (38.9% and 3.4%).

## Comparison of high perceived stress levels before and during the COVID-19 pandemic

Overall results indicate medical students who participated in the study during the COVID-19 pandemic reported a significantly higher ($p = 0.001$) prevalence of high perceived stress (20.6%) compared to students prior to the pandemic (11.6%). The same trend was observed in both male and female students (Fig. 1).

Results of the multivariable logistic regression are reported in Table 2. Factors associated with high perceived stress for all study participants, in order of highest to lowest odds ratio were; participation in the study during the COVID-19 pandemic compared to before the pandemic (aOR = 1.79, 95% CI [1.22–2.63]), female sex (aOR = 1.74, 95% CI [1.23–2.47]), younger age ($\leq 22$ years; aOR = 1.62, 95% CI [1.04–2.55]) and just sufficient income compared to sufficient and exceed (aOR= 1.53, 95% CI [1.09–2.15]).

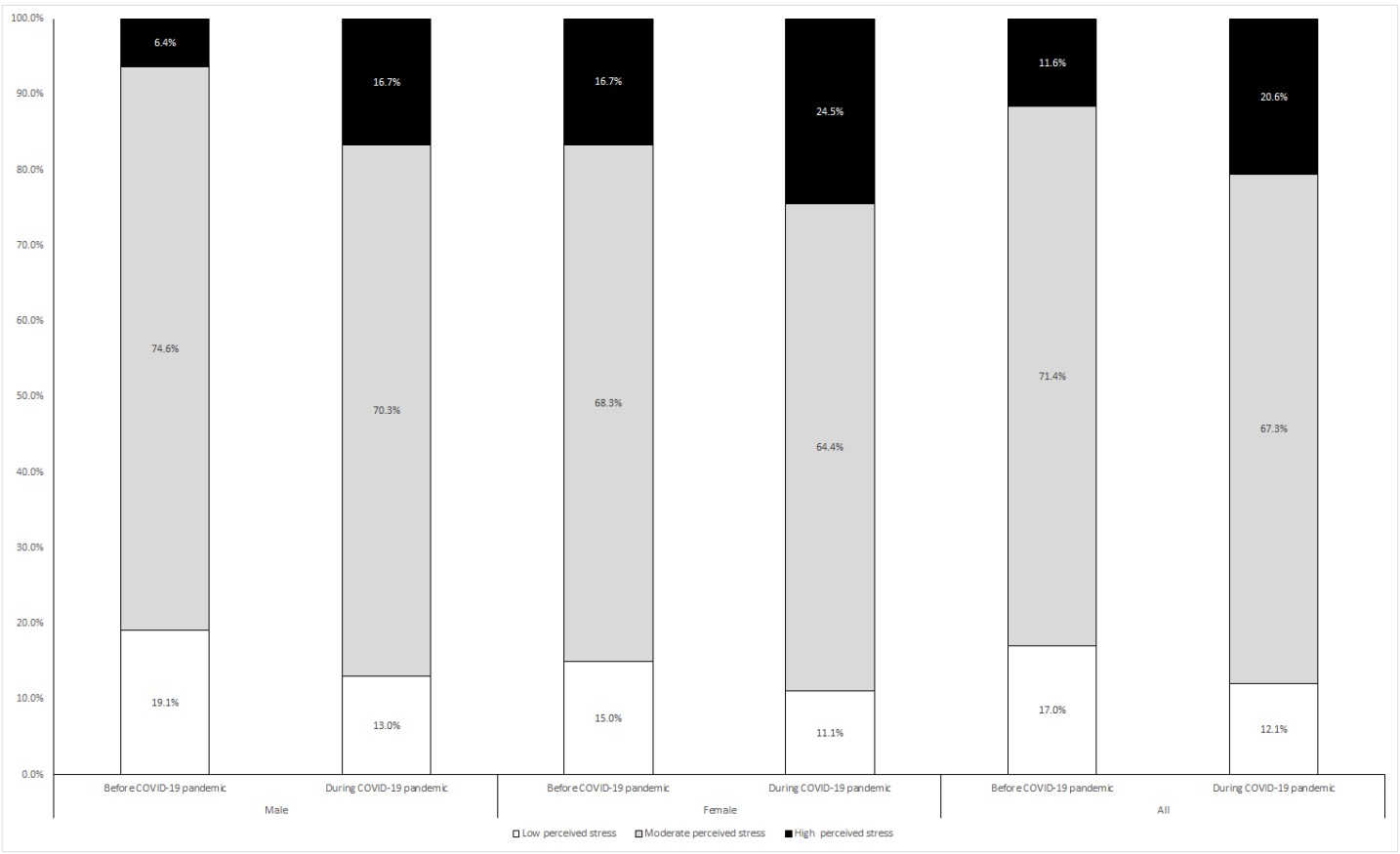

**Figure 1** Prevalence of low, moderate, and high perceived stress among male and female medical students before and during the COVID-19 pandemic.

**Table 2** Multivariable regression analysis of personal and COVID-19 pandemic-associated factors with high perceived stress among surveyed medical students ($n = 1057$).

| Factors | B | S.E. | aOR (95% CI) | P-value |
|---|---|---|---|---|
| Sex: Female vs. Male | 0.553 | 0.178 | 1.74 (1.23–2.47) | 0.002 |
| Age (years): $\leq 22$ vs. $\geq 23$ | 0.484 | 0.230 | 1.62 (1.04–2.55) | 0.045 |
| Smoking: Smoker vs. non-smoker | 0.154 | 0.301 | 0.858 (0.65–2.10) | 0.704 |
| Family income: | | | | |
| Sufficient vs. Sufficient and exceed | 0.427 | 0.173 | 1.53 (1.09–2.15) | 0.014 |
| Insufficient vs. Sufficient and exceed | 0.725 | 0.464 | 2.06 (0.83–5.13) | 0.119 |
| Timing: During pandemic vs. before pandemic | 0.584 | 0.196 | 1.79 (1.22–2.63) | 0.003 |

**Notes.**
B, unstandardized beta; S.E., standard error for B; aOR, adjusted odds ratio.

## Factors correlated with perceived stress during the COVID-19 pandemic

The average perceived stress scale (P.S.S.) score was statistically significantly ($p = 0.005$) higher among female students ($21.86 \pm 6.84$) compared to male students ($20.46 \pm 6.24$). The mean difference was $-1.41$ (95% CI $-2.38$, $-0.44$). Table 3 findings indicate that age,

**Table 3 Factors correlated with perceived stress scale score during the COVID-19 pandemic.**

| Factors | Total perceived stress score rho (*p*-value) |
|---|---|
| Age (years) | −0.144 (0.001) |
| Family income | −0.188 (0.002) |
| Academic level | −0.089 (0.004) |
| GPA | −0.023 (0.490) |
| Responses to COVID-19 related stressors | |
| How worried are you about the effect of COVID-19 pandemic on your mental health? | 0.317 (0.001) |
| How worried are you about the effects of COVID-19 virus infection for your family and friends? | 0.144.(0.001) |
| How worried are you about the effects of COVID-19 virus infection for you personally? | 0.162 (0.001) |
| How worried are you about the effects of COVID-19 pandemic on academic flow and exams? | 0.298 (0.001) |
| How worried are you about the effects of COVID-19 pandemic control measures and lockdown? | 0.200 (0.001) |
| How do you think the numerous COVID-19 pandemic news causes stress to you? | 0.220 (0.001) |
| Is the afraid of stigma and isolation due to infection causing stress to you? | 0.191 (0.001) |
| Is the uncertainty about the pandemic future and its effects causing stress to you? | 0.134 (0.001) |
| Total responses to COVID-19 related stressors | 0.302 (0.001) |

**Notes.**
Family income ranked insufficient (coded 1), sufficient (coded 2) and exceeds (coded 3), academic level (1-12).
GPA, Grade point average.
(0–5), Responses to COVID-19 related stressors ranked never (coded 0), often (coded 1), very often (coded 2).
rho, Spearman's rank correlation coefficient.

family income rank, and academic level were significantly negatively correlated with the overall P.S.S. score. Conversely, being more worried about the effects of the COVID-19 pandemic on mental health, more worried about the possible disruption of education or exams, and associated lockdown and control measures were significant factors positively correlated with P.S.S. scores (rho = 0.317, 0.298 and 0.200, respectively, $p = 0.001$). Similarly, being more worried about the effects of potential COVID-19 infection on individual health and the health of family and friends was significantly positively correlated with the P.S.S. score (rho = 0.144 and 0.162, respectively). Also, being more stressed by excessive COVID-19 pandemic news, fear of isolation and stigma due to infection and uncertainty about the pandemic future were also significant factors. The overall score for the responses to the COVID-19 related questions was significantly correlated with the P.S.S. score (rho = 0.302, $p = 0.001$).

## DISCUSSION

Concerns continue to increase about the impact of the COVID-19 pandemic and its associated control measures on medical students' perceived stress, as perceived stress

may have several adverse effects on students' mental, emotional, and physical health. Additionally, high stress may negatively impact their cognitive and learning capabilities (*Aslan, Ochnik & Çınar, 2020*).

To our knowledge, the present study is the first work to give evidence for the impact of the COVID-19 pandemic on medical students' perceived stress by using a two-part study conducted before and during the pandemic at the same college. Comparing medical students' perceived stress before and during the pandemic showed a statistically significant increase in the prevalence of high perceived stress from 11.6% pre-pandemic to 20.6% during the pandemic, indicating an augmenting effect of the pandemic on negative feelings and subjective stress among students. A similar negative impact on students' feelings was observed among Turkish (*Aslan, Ochnik & Çınar, 2020*), Polish (*Rogowska, Kuśnierz & Bokszczanin, 2020*), French (*Husky, Kovess-Masfety & Swendsen, 2020*), and Saudi (*AlHadi & Alhuwaydi, 2021*; *Alyoubi et al., 2021*) students.

Our results showed that irrespective of the timing of the study (pre or during the pandemic) females are at risk of higher stress than males. This result is consistent with studies conducted among Saudi (*Abdulghani et al., 2020*; *AlHadi & Alhuwaydi, 2021*), French (*Husky, Kovess-Masfety & Swendsen, 2020*), and Turkish (*Özdin & Bayrak Özdin, 2020*) students that also reported higher perceived stress among females. Female students may perceive challenging conditions and adverse situations as more stressful than male students (*Dyrbye et al., 2006*; *Özdin & Bayrak Özdin, 2020*).

The economic impact of the current pandemic cannot be overlooked in the present study, as students' perception of their family income as "insufficient" and just "sufficient" was significantly higher during the pandemic compared to before it. The pandemic control measures and the associated partial curfew taken in Saudi Arabia and many other countries caused many lower-income families to lose their jobs and homes (*Mumena, 2021*). Multivariable logistic regression and the correlation of the factors associated with perceived stress during the pandemic suggest that lower family income is associated with high stress. A similar result was also observed in China (*Cao et al., 2020*), Saudi Arabia (*Alkhamees et al., 2020*), and Turkey (*Aslan, Ochnik & Çınar, 2020*). One study indicated that a steady family income is protective against stress and anxiety during challenging situations, including the pandemic (*Cao et al., 2020*). To mitigate the possible financial impact of the pandemic, the Saudi government released several plans to support the private sector and people who lost their jobs due to the pandemic, including financial support and accessible health care for all (*Mumena, 2021*).

The present study indicates that being younger (≤22 years) was associated with a greater risk of high stress among students. Also, age and academic grade were negatively related to the P.S.S. score. A study conducted during the lockdown and social distancing periods revealed that younger age, female gender and lower income, were associated with a greater risk of stress and lower life satisfaction (*Rogowska, Kuśnierz & Bokszczanin, 2020*). Also, inadequate education and training about infectious diseases in the lower academic grades may increase stress and worry about the pandemic and risk of infection (*Aslan & Pekince, 2021*).

In the present study, the following COVID-19 pandemic related factors were significantly correlated with student stress levels: fear of psychological distress; worries about possible disturbances to education and exams; fear of a family member or friend getting sick; concerns about the nationwide lockdown and associated social distancing; uncertainty; fear of getting sick, being stigmatized or isolated due to infection; and exposure to excessive pandemic-related news.

A fear of psychological distress is a prominent factor for developing stress and anxiety associated with crises, especially among university students. Prompt governmental measures to contain the global crisis may have enhanced the emotional feelings of fear and distress (*Chau et al., 2019*; *Husky, Kovess-Masfety & Swendsen, 2020*).

Medical education changed from regular face-to-face teaching to e-learning methods during the pandemic. These new settings were challenging and interrupted the routine educational life of medical students. One study reported potential challenges and limitations in delivering teaching material and clinical training using e-learning technologies. Students may have been worried about inadequate clinical skill development due to insufficient clinical sessions (*Aslan & Pekince, 2021*). In the present study, worries about the possible disruption of academic lessons and exams were positively correlated with stress among medical students. These circumstances may negatively impact mental health including increasing stress and anxiety, and affect students' academic success (*Abdulghani et al., 2020*; *Aslan & Pekince, 2021*; *Charles et al., 2021*).

The present study observed that the fear of getting infected, worry about social disturbances, uncertainty, and fear that the disease may affect family members' or friends' lives were correlated with stress. Living with fear caused by the COVID-19 pandemic and mass infections is a significant source of stress and anxiety worldwide (*Abdulghani et al., 2020*; *Aslan & Pekince, 2021*; *Aslan, Ochnik & Çınar, 2020*; *Charles et al., 2021*). Universities must understand the factors that increase stress during the pandemic to improve their pandemic responses and help students cope.

The present study shows that the COVID-19 pandemic is associated with high perceived stress among medical students through two-part research done both before and during the pandemic. It also gives valuable information about personal and pandemic-related factors to help university authorities better respond to the pandemic. The results are likely generalizable to other medical students in Saudi Arabia and other countries with similar COVID-19 situations. However, this study has some limitations. One, it was a cross-sectional study, which minimizes the ability to prove causal relationships; a cohort design would be a stronger study type for establishing the impact of the COVID-19 pandemic on perceived stress. Also, perceived stress levels may not be enough to represent the mental health status of the students. Another limitation is the use of a non-random sample which may limit the generalizability of the results. Finally, there is a possible reporting bias due to online self-reported questionnaires in the second part of the study; respondents may have reported answers that they thought were anticipated rather than answers that reflected their reality.

## CONCLUSIONS

This study provides evidence that the COVID-19 pandemic increased the level of stress among medical students. Younger students, those in lower academic levels, and female students are at risk of having high stress. Worries about possible disruption to academic progress or exams, exposure to excessive news, worries about the possibility of COVID-19 infection, and the effects of mandatory isolation and social distancing were important sources of stress. The implementation of online stress management programs is recommended to improve stress management and coping strategies among university students. Particular emphasis should be given to females, younger students, and students in low academic grades. To help students fight their fear, they should be given accurate and recent information about the ongoing pandemic. Online training, especially for students in low academic grades, may be necessary to improve their infection control practices. Measures should be taken to improve online teaching methods. A survey about preferable exam methods during the pandemic may be necessary.

## ACKNOWLEDGEMENTS

The authors would like to acknowledge all the participating medical students for spending their valuable time completing the questionnaire for this study.

### Funding

This work was funded by the Institute of Research and Consulting Studies at King Khalid University (grant number # 4-N-20/21). The funders had no role in study design, data collection and analysis, decision to publish, or preparation of the manuscript.

### Grant Disclosures

The following grant information was disclosed by the authors:
Institute of Research and Consulting Studies at King Khalid University: #4-N-20/21.

### Competing Interests

The authors declare there are no competing interests.

### Author Contributions

- Nabil J. Awadalla conceived and designed the experiments, performed the experiments, analyzed the data, prepared figures and/or tables, authored or reviewed drafts of the article, students data collector training, and approved the final draft.
- Abdullah A. Alsabaani performed the experiments, authored or reviewed drafts of the article, and approved the final draft.
- Mohammed A. Alsaleem performed the experiments, authored or reviewed drafts of the article, and approved the final draft.
- Safar A. Alsaleem performed the experiments, authored or reviewed drafts of the article, and approved the final draft.

- Ayoub A. Alshaikh performed the experiments, authored or reviewed drafts of the article, and approved the final draft.
- Suliman H. Al-Fifi performed the experiments, authored or reviewed drafts of the article, and approved the final draft.
- Ahmed A. Mahfouz conceived and designed the experiments, performed the experiments, analyzed the data, prepared figures and/or tables, authored or reviewed drafts of the article, and approved the final draft.

### Human Ethics

The following information was supplied relating to ethical approvals (i.e., approving body and any reference numbers):

Ethical Committee of the Scientific Research, King Khalid University, has approved (ECM#2019-23) the first and the second (ECM#2020-1205) surveys. The participants provided their written informed consent to participate in this study.

### Data Availability

The raw data is available in the Supplemental Files.

### Supplemental Information

Supplemental information for this article can be found online at http://dx.doi.org/10.7717/peerj.13900#supplemental-information.

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
