# Peer review of "Increased mental stress among undergraduate medical students in south-western Saudi Arabia during the COVID-19 pandemic"

_PeerJ, doi:10.7717/peerj.13900_

## Round 0.1 · original submission · Major Revisions

Thank you for submitting the manuscript to PeerJ. It has been reviewed by experts in the field and we request that you make major revisions before it is processed further.

We look forward to hearing from you soon.

Best wishes,

Badicu Georgian, Ph.D

Reviewer 1 ·

Basic reporting

This is an interesting area of research however the literature review has not been carried out in detail as there are a number of studies that have been published on this topic. The introduction does not provide a sufficient critical approach either.

Experimental design

Very limited when considering methodological approach.

Validity of the findings

I am not sure if there are any novel findings since some of the recent papers have already provided a good amount of data from Saudi Arabia. Thus I would like to encourage the team to review current studies and then collect more data and carry out more sophisticated analyses of the data.

Reviewer 2 ·

Basic reporting

First of all, I would like to thank you for the opportunity to review the paper entitled: Increased mental stress among undergraduate medical students in south-western Saudi Arabia during the COVID-19 pandemic. In this research, authors evaluate the stress among undergraduate medical students during the Covid -19 Pandemic. In general, the paper has an interesting and relevant topic for researchers and medical professors and the written level of English is very good and easy to read. However, the methodology and main results need to be rewritten. With your permission I recommend the following:
Some information are confusing:
The information about first survey - what are the questions and if also the Cohen questionnaire was completed? We need more information about this two topics.
In the second survey, the number of the students is almost double comparing with first survey? Did you collect data from the same students in different periods or from different students in different moments? If your data are not collected from the same students, how you will control the stress level?

Experimental design

Line 191 - T test was used to compare the P.S.S, there is any table with t values?
It was difficult to understand why did you use a multivariable analysis and after a correlation, there is any arguments for using this methods?
For a better understanding please insert the statistical description for each variable in a table. Please, insert in a table all the results from the questionnaire and after the table results comparing both sessions, including what was the statistical test that you use, the T test or Mann Whitney U test and effect size for each comparison.

Validity of the findings

Your work was impressive and after the researchers from the field, the second actor are the professors. Please follow the needs and describe your work for both size (for researchers insert all the details for replicate the study with other representative samples and for professors - describe more clearly the technique that you used in table 2 and 3.

Additional comments

No additional comments

---

## Round 0.2 · Major Revisions

Thank you for submitting the manuscript to PeerJ. It has been reviewed by experts in the field and we request that you make again major revisions before it is processed further.

We look forward to hearing from you soon.

Best wishes,

Badicu Georgian, Ph.D

Reviewer 2 ·

Basic reporting

Congratulations on your work in answering the questions. After rereading the manuscript, there are many more doubts about the presentation of your data, especially the logistic regression and the spearman correlation. Unfortunately, even if the number of participants is large (1074 students), the results are confusing. For logistic regression, you used multinominal logistic regression, if I understood correctly, please provide us with more information about the model. There is an example how to report the results Andy Field book (Discovering statistics using SPSS) page 284-312. I was trying to understand your results, but I'm still confused.
Regarding the table with the correlations Spearmon (rho), the results are very poor, correlation spearman = 0.14, p.001? I think this is a mistake. Thus, please redo the results section, indicating for each variable used, including the stress questionnaire, the description of each variable, and then argue the use of each analysis in order to understand the results. Your work is extraordinary, but the results are not clear.

Experimental design

The experimental design is well describe.

Validity of the findings

A detailed description of the approach is needed. We do not have enough data on each variable. Another relevant aspect concerns the reasoning of each analysis, why these analyzes were used and what the impact is on the results. Please follow the steps in reporting the results.

·

Basic reporting

Overall, I found the paper well written and professionally and appropriately structured. There a few very minor edits required to the language.

Experimental design

This is a cross-sectional and therefore observational study. The aims of the research were clearly stated and appropriately placed at the end of the Introduction. The analyses were appropriate for the study. As the data were provided, I did obtain some slightly different results than the authors. In general the authors provided sufficient information to allow for replication of the results except in those instances referred to in this review, when it is not clear why some slightly different coefficients were obtained in my analysis. I apologise if I am in error.

Validity of the findings

Overall I support the findings reported by the authors, with the exception of a few issues mentioned in this review.

Additional comments

INTRODUCTION
(with line numbering beginning on P1 of the review pdf)
Line 64: ‘Although young adults are at low risk of COVID-19 infection (Vieira et al. 2020), college students may represent a vulnerable population for mental health problems during the pandemic’. This may have been the case with the original strain of Covid19, when the study was done, but is not necessarily the case now. I suggest some qualifying statement perhaps ‘Although young adults are at low risk of COVID-19 infection from the original strain of the virus (Vieira et al. 2020)’, or perhaps raise this point in the Discussion.
Line 75 – 77: ‘Therefore, we hypothesize that COVID-19-related challenges would represent a more considerable stressor and result in a more significant increase in perceived stress during the pandemic’.
I suggest deleting the word ‘more’ from this sentence (both times it occurs), otherwise the reader is left asking ‘more than what’?
Line 95: ‘…exceeds the pre-pandemic prevalence among the same college students’. This is not entirely the case as the current study did not use the same college students; rather the current study used similar college students.
STUDY PARTICIPANTS AND PROCEDURE
For both the first and second surveys, were the students a mix of years, eg 1st, 2nd, 3rd etc? All we are told is that they are undergraduates.
Line 112: What is meant by ‘level eight’ medical students? Please clarify what is meant by ‘academic level’.
Line 128: It would be helpful to know the denominators here: 353 out of how many, and 704 out of how many?
Line 134: ‘Students were stratified by academic level with a cluster chosen from each level’. So was the academic level the cluster? Also, the way the sentence is written makes it sound as though there could have been a number of clusters for each level, out of which one cluster was selected. The next sentence says ‘All listed students in each group were involved in the study whenever possible’. Is a group the same as a cluster?
Line 133: Currently reads ‘In the second one…’, suggest changing to ‘In the second survey…’, but once again, please clarify what is meant by ‘level.’
STUDY TOOL
Line 179: I understand word count limitations, but please briefly explain how the eight Covid questions were validated, e.g., how many experts, what process was followed, etc. Also, Cronbach’s alpha specifically measures internal consistency reliability, so I suggest ‘…and Cronbach’s alpha was 0.86, indicating internal consistency reliability of the scale’ (2 decimal places here are sufficient).
DATA ANALYSIS
Line 194: Please cite SPSS as IBM SPSS Statistics version 22.
Line 195 – 199: Please re-write as ‘Chi square test of independence was used to compare the personal and academic data of the students before and during the COVID-19 pandemic. A multivariable logistic regression analysis model was used to assess the independent factors of high perceived stress in the whole sample.’
RESULTS
Line 210: Please re-write as follows ‘The two groups were matched on age, sex etc’. Next sentence, beginning ‘However…’, change to ‘…the family income of the two groups was statistically significantly different’.
Line 212: In text here, the income variable is reported as ‘satisfactory’ or ‘unsatisfactory’ but in Table 1 the levels are ‘Sufficient and exceed’, ‘Sufficient’ and ‘Insufficient’. Check Table 2 also for this variable. Please use the same terminology between tables and text. Was the difference between these two levels determined by obtaining the Adjusted Standardised Residuals from the Crosstabs procedure in SPSS?
The Age variable is reported as <=22 and +22. For clarity and to avoid confusion, I suggest reporting the +22 as => 23. This also applies to Table 2. What was the justification for dichotomising the age variable at 22? Logistic regression can have continuous variables as predictors.
Figure 1: was this analysed using Chi Square? This is not in the Data Analysis section unless it comes under the ‘personal’ information.
Line 223: I suggest re-writing as ‘Results of the multivariable logistic regression are reported in Table 2. Factors associated with high perceived stress in order of highest to lowest odds ratio were; etc etc. It would be helpful to include the p value in table 2 as per table 2, rather than just asterisk and say ‘Statistically significant’.
It is so helpful to have the data available. I ran the multivariable logistic regression but obtained slightly different results. Apologies if I have made a mistake here. Did the authors include any other variables here that are not in Table 2? Overall interpretation is the same except for Smoking, where in my analysis the OR was <1 (but still non-significant) and Income, which was not significant.
Was there any reason why the dichotomised GPA and Level variables were not included?
Line 229: Heading for this section – the word ‘stress’ needs to be inserted after ‘perceived’.
Line 230: It looks like a t-test has been performed here between male and female PSS scores but there is no mention of this in the Data Analysis. I’m assuming the means and standard deviations are reported. If a P value is <0.05, please refer to it as ‘statistically significant’ rather than just ‘significant’. While the male and female PSS scores do differ, the effect size (Cohen’s d) between male and female means – 0.215, or small. Does a small effect size here mean anything? The authors have a reasonable sample size here, and so relatively trivial differences can be statistically significant but of no clinical or practical importance.
Likewise most of the correlations in Table 3 are also small (0.10 = small, 0.30 = medium, 0.50 = large). The effect of the pandemic on health and academic flow and exams are the 2 strongest, the others are relatively weak. The authors said in the data Analysis section that non-parametric correlations were used, so I’m assuming Spearman’s? Please note – Spearman’s coefficient is not known as r (that is for Pearson’s correlation coefficient. The term ‘r’ is also used in line 237, 240 & 244). Spearman’s is known as rho. In Table 3, is the P value for GPA correctly reported or is it the asterisk that is incorrect? In the table the P value is reported as 0.543, but is asterisked as statistically significant. By the size of the correlation (-0.023), I’m guessing it is not statistically significant. It is also not necessary to asterisk the P values and then indicate below the table in text that these are statistically significant – a reader should be able to determine than anything <0.05 is statistically significant.
Please also check the wording of the last 2 questions in Table 3.
Based on the supplied data, I ran the correlations between total PSS score and GPA, and total PSS score and level, and obtained different results. Please see attachment.
DISCUSSION
Line 254: Comparing medical students' perceived stress before and during the pandemic showed a significant increase in high perceived stress from 11.6% pre-pandemic to 20.6% during the pandemic.
Please re-write as Comparing medical students' perceived stress before and during the pandemic showed a statistically significant increase in the prevalence of high perceived stress from 11.6% pre-pandemic to 20.6% during the pandemic.
Line 262: Moreover, multivariable logistic regression demonstrated that female students were at a higher risk of high stress during the pandemic than males.
This isn’t actually the case. The regression showed that females were at an increased risk of high perceived risk compared to males, regardless of the timing.
Line 275: Multivariable logistic regression and the correlation of the factors associated with perceived stress during the pandemic suggest that lower family income is significant in high stress.
Two points here: 1) which variable in Table 3 indicates a relation ship between income and stress? I don’t see income specifically mentioned. 2) Re-write the last part of the above sentence as ‘…lower family income is associated with high stress’.
Line 287: A study conducted during the lockdown and social distancing periods revealed that younger age, besides the female gender and lower income, is correlated with a greater risk of stress and lower life satisfaction (Rogowska et al. 2020).
Please re-write as ‘A study conducted during the lockdown and social distancing periods revealed that younger age, female gender and lower income, were associated with a greater risk of stress and lower life satisfaction (Rogowska et al. 2020).
Line 290: t is assumed that the ability of younger students to cope with stress is not entirely mature.
It is not clear what this sentence means. Do the authors mean that due to younger age, students may not necessarily have the maturity to cope with stress?
Line 307: One study reported potential challenges and limitations in delivering teaching material and clinical training using e-learning technologies. Please provide the reference for this study.
CONCLUSIONS
Line 341: Younger students, those in lower academic grades, and female students are at risk of having high stress.
The only analysis I can see involving grades (GPA) is in Table 3, and here the association between GPA and stress is not statistically significant. I attempted to replicate these results (see attachment) and obtained a somewhat different, but statistically non-significant correlation between GPA and stress. I therefore do not see any evidence that lower grades are associated with stress. Academic level also only appears in the correlations (and is statistically significantly negatively associated with stress) but the relationship is very weak.

---

## Round 0.3 · Major Revisions

Thank you for submitting the manuscript to PeerJ. It has been reviewed by experts in the field and we request that you make again major revisions before it is processed further.

We look forward to hearing from you soon.

Best wishes,

Badicu Georgian, Ph.D

·

Basic reporting

No comment

Experimental design

No comment

Validity of the findings

No comment

Additional comments

GENERAL
I would like to congratulate the authors and responding to the reviewer’s comments and making the paper an improved piece of work.
I have made a number of comments and suggested edits, plus a few things I missed on my first review, apologies. These are mainly relatively minor. I did identify that the ‘Income’ variable has not been properly handled in the regression, but this does not affect the interpretation of results in any major way.

INTRODUCTION
2nd paragraph, line 59: recommend deleting the comma after ‘decreased life satisfaction’ and changing to ‘decreased life satisfaction and decreased positive emotions’.
7th paragraph, line 96: delete ‘the’ between ‘among’ and ‘similar’.
METHODS
Study tool
Line 183: ‘…after an extensive review of the literature, a panel of four experts in the…’ delete the comma after literature and insert the word ‘and’.
Line 185: Please state the method for assessing the content validity – was it the Content Validity Ratio (CVR)?
Line 185: “The Cronbach’s Alpha…’ please delete ‘The’, and Alpha can be lowercase (alpha).
Data analysis
Line 200: Please insert ‘Statistics’ after IBM SPSS.
RESULTS
Comparison of high perceived stress levels before and during the COVID-19 pandemic
I now see what has happened with the Income variable. Unfortunately, you have not used the Income variable correctly in the logistic regression. You have treated it as if it is a continuous variable (1, 2, 3) but it is an ordinal variable, and must be used in this way in the logistic regression, otherwise the coefficients are not correct. The text (lines 209 – 213) says all predictor variables were dichotomised, but this is not the case for income.
STEP 1: Include the 3 level ‘Income variable’ in the list. Click on ‘Categorical’

STEP 2: Click on ‘Income’ and make sure ‘First’ is selected as reference category. This is the only appropriate reference category as the last (Insufficient) has too small a sample size to be a reference for which the other 2 levels are compared.

The test will compare ‘Sufficient and exceeds’ to ‘Insufficient’ and ‘Sufficient and exceeds’ to ‘Insufficient’, and you will then have 2 sets of statistics – B coefficients, SEs, aORs and P values, for this variable.
This is what I get if I use the 3-level income variable

The reference is ‘Sufficient and exceeds’ here, and Income 1 = Sufficient, Income 2 = Insufficient’.
So ‘Sufficient’ vs ‘Sufficient and exceeds’ is statistically significant (P = .014). Compared to students with ‘Sufficient and exceeds’ income, those with ‘Sufficient’ are more likely to experience high stress (aOR 1.53 (1.09, 2.15) but ‘Insufficient’ vs ‘Sufficient and exceeds’ is not statistically significant (P = .119). This is probably due to the sample size in ‘Insufficient’ (n = 28).
If I run the logistic regression WITHOUT specifying Income as a categorical variable, so that the test thinks it is a continuous variable, I get the same results as your table 2. See below.

In your Data analysis section, I recommend writing this as
‘The following predictors of high perceived stress were dichotomized variables: sex (female vs. male), age in years (≤22 vs. ≥23), smoking status (smoker vs. non-smoker) and the timing of the study (during the pandemic vs. before the pandemic). Income was a 3-level variable as per Table 1, with ‘Sufficient and exceed’ as the reference category.
I understand this changes your argument somewhat regarding the regression, but it is not appropriate to treat an ordinal variable as if it were continuous, as treating it as continuous assumes that the difference in stress between level 1 and 2 is the same as the difference in stress between 2 & 3 and you can’t make that assumption.
You can use the rankings for Income in the correlation because you have used Spearman’s, but there is no equivalent in the logistic regression.
Also please note that the symbol β is for a standardised beta, whereas in the logistic regression it is unstandardised, and so should just be reported as B.
Line 215: It would be helpful to indicate here the scoring for family income, e.g., ‘income ranked insufficient (coded 3), sufficient (coded 2) and exceeds (coded 1); educational level (1 – 12) and GPA (0 – 7).’ If you don’t want to put them here, I suggest putting them below Table 3, as this helps the reader understand the correlations and they don’t have to remember or read back over previous tables/text. Also helpful to have rankings for the 8 Covid questions provided, ie Never = 0, Very often = 2).
Factors correlated with perceived stress during the COVID-19 pandemic.
It is appropriate to provide some measure of effect size in any analysis. I recommend adding in the following.
The average perceived stress scale (P.S.S.) score was statistically significantly (p = 0.005) higher among female students (21.86 ± 6.84) compared to male students (20.46 ± 6.24). The mean difference was -1.41 (95% CI -2.38, -.44).
I note that although there is statistical significance, both means are in the ‘moderate range’.
Line 249: Please remove ‘increased’, you only need to say age here, as the negative correlation implies that as people get older, stress decreases.
Line 252: should be ‘worried’ not ‘worries’.
DISCUSSION
Line 274: There is a fullstop after ‘pandemic’ that needs to be removed.
Lines 279 – 282: Our results showed that high stress was more prevalent among female students compared to male students both before and during the pandemic. Moreover, multivariable logistic regression verified that female students were at a higher risk of high stress.
The t test reported in lines 248-249 showed that female mean stress higher than males during the pandemic. Note - means do not indicate prevalence. No t-test of means between males and females pre-pandemic was reported. The regression showed that, irrespective of pre or during, females are more at risk of higher stress. I therefore think you do not need both these statements – just a statement that irrespective of the timing of the study (pre or during the pandemic) females are at risk of higher stress than males.
Line 308: It is assumed that the younger students face difficulties coping with stressful conditions.
This is a big assumption to make. I also think it is unnecessary, as the preceding sentence and the Rogowska reference covers this issue.
Line 318: A fear of psychological distress is a prominent factor for developing stress and anxiety associated with crises, especially among university students. Do the Chau et al and Huskey et al include this information about university students? Otherwise a reference is needed for this.
Line 328: In the present study, worries about the possible disruption of academic lessons and exams were a major source of stress among medical students.
Remember, correlation is not causation! There was moderate correlation between stress and academic disruption, so all you can say is
In the present study, worries about the possible disruption of academic lessons and exams were positively correlated with stress among medical students.
Likewise in line 335 – replaces ‘sources of stress’ with correlated with stress.

---

## Round 0.4 · accepted · Accept

Thank you for making the final revisions to your manuscript.